# The effect of soil sample size, for practical DNA extraction, on soil microbial diversity in different taxonomic ranks

Hiroki Morita◉, Satoshi Akao◉*

Graduate School of Science and Engineering, Doshisha University, Kyotanabe, Kyoto, Japan

* sakao@mail.doshisha.ac.jp

## Abstract

To determine the optimal soil sample size for microbial community structure analysis, DNA extraction, microbial composition analysis, and diversity assessments were performed using soil sample sizes of 0.2, 1, and 5 g. This study focused on the relationship between soil amount and DNA extraction container volume and the alteration in microbial composition at different taxonomic ranks (order, class, and phylum). Horizontal (0.2 and 1 g) and vertical (5 g) shaking were applied during DNA extraction for practical use in a small laboratory. In the case of the 5 g soil sample, DNA extraction efficiency and the value of α-diversity index fluctuated severely, possibly because of vertical shaking. Regarding the 0.2 and 1 g soil samples, the number of taxa, Shannon–Wiener index, and Bray–Curtis dissimilarity were stable and had approximately the same values at each taxonomic rank. However, nonmetric multidimensional scaling showed that the microbial compositions of these two sample sizes were different. The higher relative abundance of taxa in the case of the 0.2 g soil sample might indicate that cell wall compositions differentiated the microbial community structures in these two sample sizes due to high shear stress tolerance. The soil sample size and tube volume affected the estimated microbial community structure. A soil sample size of 0.2 g would be preferable to the other sample sizes because of the possible higher shearing force for DNA extraction and lower experimental costs due to smaller amounts of consumables. When the taxonomic rank was changed from order to phylum, some minor taxa identified at the order rank were integrated into major taxa at the phylum rank. The integration affected the value of the β-diversity index; therefore, the microbial community structure analysis, reproducibility of structures, diversity assessment, and detection of minor taxa would be influenced by the taxonomic rank applied.

## Introduction

The soil environment possesses abundant microorganisms with $10^{10}$–$10^{11}$ bacteria and 6,000–50,000 species present in 1 g of soil [1]. Microbial composition and diversity in the soil environment interact with various biotic and abiotic factors [2]. Moreover, soil microorganisms are not uniformly distributed because of the physical and chemical characteristics of the environment. Nunan et al. [3] indicated that the occurrence of different bacterial distributions

**Data Availability Statement:** database URL: https://www.ddbj.nig.ac.jp/index.html accession number:DRA011569.

**Funding:** This study is supported by "The Sanitation Value Chain: Designing Sanitation

Systems as Eco-Community Value System" Project (Project Leader: Prof. Taro Yamauchi) at Research Institute for Humanity and Nature (RIHN, Project No. 14200107). The funder had no role in study design, data collection, analysis, decision to publish, nor preparation of the manuscript.

**Competing interests:** The authors have declared that no competing interests exist.

depends on the nutrients in the soil. The distribution of bacteria in soil is known to be affected by pH and water content [4, 5]. When analyzing the microbiome in the soil, it is necessary to apply reliable sampling methods considering the heterogeneity of bacterial distribution. The first possible solution to this problem is to use a large number of mixed soil cores. This allows us to obtain representative and reliable results for the entire soil sampling area [6, 7], although it is time-consuming and costly.

Soil sample size, i.e., sample volume, is another factor that determines the reproducibility of the soil microbial diversity analysis. Several studies have reported the effect of soil sample size on microbial diversity analysis using the polymerase chain reaction-denaturing gradient gel electrophoresis (PCR-DGGE) method. In the case of a large soil sample size (e.g., 10 g of soil), the dominant taxa accounted for a larger bacterial population ratio than that in the case of a smaller soil sample size, although the large soil sample case had larger variances in DNA extraction efficiency and microbial community structure [8]. In the case of a small soil sample size (e.g., 0.25 g), minor species could be detected, and the reproducibility of the bacterial community was high [8]. However, using small soil sample sizes (e.g., 0.1 g), another study reported high variability in bacterial structure among samples, with the detection of minor species [9]. As a result, these reports using the PCR-DGGE method do not provide a standard for the appropriate soil sample size for microbial diversity analysis.

On the other hand, amplicon sequencing analysis provides a different perspective [10]. In the reported analysis, a soil sample size of 10 g, the largest soil volume for a certain DNA extraction kit used in the study with 1, 5, and 10 g of soil sample sizes, presented the lowest variability among the repeated microbial diversity analyses and the highest microbial community reproducibility. Furthermore, minor taxa could be stably detected even in the case of the largest soil volume because of the large number of sequence reads, with an average of over 56,000 reads. Song et al. [11] also investigated the effect of soil sample size (0.25 and 10 g) on fungal amplicon sequence data.

Although the study by Penton et al. [10] was insightful, in our opinion, it can benefit from further clarifications, such as those concerning target taxonomic rank and annotated taxa in the microbial diversity analysis, the relationship between soil volume and tube volume for DNA extraction, and DNA extraction cost and efficiency. First, the focused taxonomic rank and related annotated taxa might affect microbial diversity analysis. Penton et al. [10] did not describe the targeted taxonomic rank nor the annotated taxa used for microbial diversity analysis. Regarding the taxonomic rank, a taxon belonging to a minor category in a lower taxonomic rank (e.g., order) might be integrated into a major category in an upper taxonomic rank (e.g., phylum). In other words, rare taxa in lower taxonomic ranks might not be able to be classified or recognized as minor taxa in the upper taxonomic ranks. Thus, the recognition of rare taxa would depend on the target taxonomic rank. For the annotated taxa, there would usually be numerous operational taxonomic units (OTUs) not belonging to any taxa in an upper taxonomic rank (e.g., phylum) from amplicon sequence data. If these OTUs are counted as many independent pseud-taxa, the microbial diversity analysis would be biased by these OTUs. Therefore, we think that the target taxonomic rank and the annotated taxa used should be clearly defined. Second, the relationship between soil volume, reagent volume, and tube volume may affect the efficiency and quality of DNA extraction. Penton et al. [10] used the PowerMax Soil DNA Isolation Kit (QIAGEN, Venlo, Netherlands) with a 50 mL tube for DNA extraction from 1 to 10 g of soil. They also used the PowerSoil DNA Isolation Kit (QIAGEN) with a 2 mL tube for DNA extraction from 0.25 g of soil. The performance of amplicon sequence analysis using 0.25 g of soil was better than that using 1 g of soil, the lowest ratio of soil volume to container volume (50 mL), based on the variability and reproducibility of the results. Song et al. [11] also used each kit for DNA extraction from 10 g and 0.25 g of soil,

respectively. There was no significant difference in OTU richness between the two soil sample sizes, possibly because of the appropriate addition of soil volume into each kit. Finally, using a 50 mL tube for DNA extraction is expensive and inefficient; the number of tubes used for the extraction at one instance is limited if we use the recommended attachment providing horizontal vibration (Vortex Adapter for two tubes (50 mL), QIAGEN) for the extraction kit. This might be an obstacle to the processing of large amounts of samples. One possible alternative to bypass the one-time sample size (number of tubes) limitation is to use a device with vertical shaking, rather than with horizontal shaking.

In this study, we investigated the effects of changes in soil sample volume on microbial diversity using amplicon sequence analysis. This could be considered in accordance with two factors: a change in tube volume preferentially in parallel to the change in soil sample volume, and the application of a device for vertical shaking intended for DNA extraction. Microbial community structure analysis at different taxonomic ranks using the annotated taxa from the amplicon sequence data was also performed to evaluate the effect of the taxonomic rank applied on the analysis.

## Materials and methods

### Soil sampling and DNA extraction

Soil samples were collected from a local field for vegetable crops (34˚41.32N, 135˚49.91E). The collected soil was mixed well and passed through a 2 mm sieve. Five samples (n = 5) of three different soil sample sizes (0.2, 1, and 5 g) were prepared for each sample. The number of repetitions per sample (n = 5) was determined based on previous reports [7, 11]. DNA extraction was performed in a centrifugal tube using the phenol/chloroform/isoamyl alcohol (PCI, 25:24:1) reagent (Nippon Gene, Tokyo, Japan). To prevent DNA degradation and adsorption by humic acid, glass beads with a diameter of 1.5–2.5 mm (BZ-02, AS ONE, Osaka, Japan) and skim milk (FUJIFILM Wako Chemicals, Osaka, Japan) were added to the tube. The tube was shaken at high speed for 5 min using a mixer with its attachment (Table 1). Recovery of the supernatant (water phase) containing the extracted DNA was performed twice in each tube with a 2-fold addition of the PCI reagent. The specifications of DNA extraction in each case, such as tube volumes, reagents, and additives, are summarized in Table 1. Extracted DNA was collected from the supernatant after centrifugation at $10,000 \times g$ for 2 min (CF15RXII, Hitachi Koki, Tokyo, Japan), and the DNA was purified by ethanol precipitation. The concentration and quality of the recovered DNA were measured using the fluorometer (Qubit 4, Thermo

**Table 1. DNA extraction conditions.**

| Soil sample size (g) | 0.2 | 1 | 5 |
|---|---|---|---|
| Tube size (mL) | 2 | 15 | 50 |
| Glass beads (g) | 0.02 | 0.1 | 0.5 |
| Skim milk (mg) | 2 | 10 | 50 |
| PCI reagent (mL) | 0.2 | 1 | 5 |
| Mixer for Beat Beating | Voltex Genie 2 (Scientific Industries, New York, USA; rated power output, 0.1 kW) | | ASCM-1 (AZ ONE; rated power output, 0.05 kW) |
| Attachment for Mixer | Horizontal Plastic Clip Microtube Holder (Scientific Industries; operational number of tubes, 24) | Horizontal 15 mL Tube Holder (Scientific Industries; operational number of tubes, 12) | ASCM-R50 (AZ ONE; operational number of tubes, 6) |

The shaking strength of the mixer was set to approximately 2,700 rpm (Voltex Genie 2) and 2,000 rpm (ASCM-1).

Fisher Scientific) and the Qubit dsDNA BR Assay Kit (Thermo Fisher Scientific). All reagents used were of analytical or biochemical grade.

### 16S rRNA gene amplification and amplicon sequencing

The 16S rRNA gene was amplified using a primer set (V3V4f_MIX and V3V4r_MIX) [12] and a thermal cycler (MyCycler TM 170-9703JB, BIO-RAD, Hercules, USA). Bovine serum albumin (2.5 µL) (Takara Bio, Kusatsu, Japan) was added to prevent PCR inhibition. Amplicon sequencing was performed by an external analytical laboratory (Bioengineering Lab., Sagamihara, Japan, https://gikenbio.com/) according to the manufacturer's protocol. In brief, high-throughput sequencing of amplicons was performed using the Illumina MiSeq platform ($2 \times 300$ bp). Raw paired-end reads were subjected to quality filtering using the Fastx toolkit. After primer sequence removal, the sequences were analyzed using QIIME 2.0. The EzBioCloud 16S database (version 20200225) was used to detect OTUs with 97% similarity [13].

### Microbial community structure analysis

Raw OTUs, which were estimated to be a certain valid name in the order rank, were categorized into relative abundance at the order rank. Five amplicon sequencing results were obtained for each soil sample size. The variance in microbial diversity among the soil sample sizes was compared based on the relative abundance at the order rank. Following the previous studies, the Shannon–Wiener index (H′) was applied as the alpha diversity index, and Bray–Curtis dissimilarity and Sørensen–Dice indices were applied as the beta diversity indices [14, 15]. The same indices were applied at the class and phylum ranks to elucidate the possible variance in diversity assessment.

### Statistical analysis

Significant differences in DNA extraction efficiency were examined by multiple comparisons using the Bonferroni method. ANOVA was performed on the average number of non-chimeric sequence reads for each soil sample size (0.2, 1, and 5 g). Parametric *t*-tests were performed between the average number of OTUs derived from the 0.2 and 1 g samples. Wilcoxon rank-sum test was used to evaluate the difference in each diversity index from the 0.2 and 1 g samples. Non-metric multidimensional scaling (NMDS) based on Bray–Curtis dissimilarity was applied to compare the cohesion of plots across the soil sample sizes. Linear discriminant analysis was performed between the average numbers of OTUs derived from the 0.2 and 1 g samples using LEfSE [16]. All statistical analyses were performed using R (ver 3.6.1) with the "vegan" and "MASS" packages.

## Results and discussion

### DNA extraction

The DNA extraction efficiency is shown in Fig 1. It was calculated by dividing the measured DNA concentration (µg mL$^{-1}$) by the soil sample size (g). Ideally, DNA extraction efficiency should not change with a change in soil sample size. However, the extraction efficiency corresponding to the soil sample size of 1 g was significantly higher ($\alpha < 0.05$) than that of the other two sample sizes, based on the multiple comparisons performed.

The low extraction efficiency with a sample size of 0.2 g might be owing to the DNA extraction process. The supernatant was recovered from the DNA extraction tube, and small but unrecoverable portions were always generated, especially with the 0.2 g sample using a small amount of the extraction reagent, to avoid taking an intermediate layer containing proteins. In

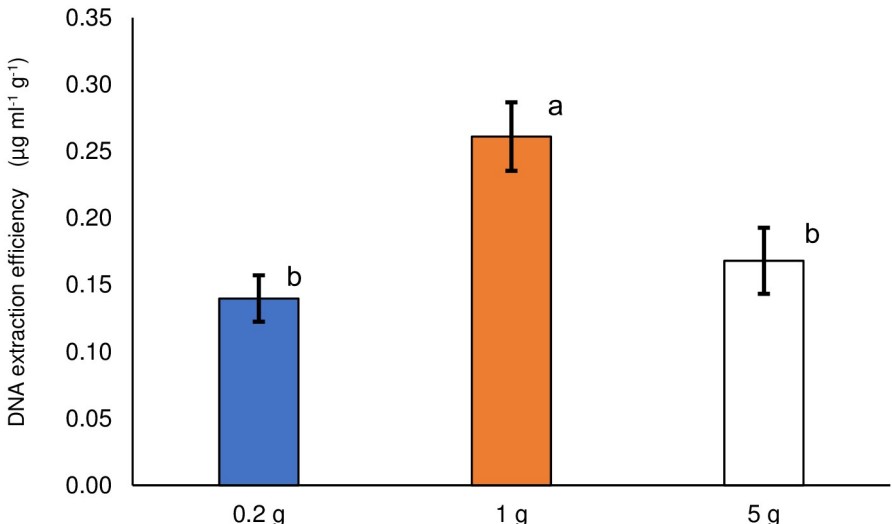

**Fig 1. DNA extraction efficiency by soil sample size.** Different lowercase letters indicate significant differences ($\alpha <$ 0.05). Error bars represent standard errors.

contrast, the low extraction efficiency with a sample size of 5 g might be because of the bead-beating strength. DNA extraction for the 0.2 and 1 g soil samples was performed with horizontal shaking, whereas vertical shaking was used in the case of the 5 g soil sample. This might have resulted in the low efficiencies of cell disruption and DNA extraction.

## Amplicon sequencing

Non-chimera reads, total OTUs, and non-single/double OTU sequences for each sample are shown in Table 2. After quality filtering and chimera removal, we obtained 604,543 reads, with a range of 29,043–45,888 reads per sample. These numbers are adequate for achieving

**Table 2. Number of reads, OTUs, and OTUs excluding singleton and doubleton in each sample.**

| Sample name | Non-chimera reads | Total OTUs | Non-S/D OTUs |
| --- | --- | --- | --- |
| soil-02g-1 | 39220 | 1959 | 1909 |
| soil-02g-2 | 42242 | 1641 | 1612 |
| soil-02g-3 | 45395 | 1733 | 1694 |
| soil-02g-4 | 45888 | 1910 | 1895 |
| soil-02g-5 | 44782 | 1653 | 1604 |
| soil-1g-1 | 39646 | 1809 | 1744 |
| soil-1g-2 | 35511 | 1709 | 1651 |
| soil-1g-3 | 35930 | 1574 | 1519 |
| soil-1g-4 | 41860 | 2029 | 1971 |
| soil-1g-5 | 44150 | 1911 | 1852 |
| soil-5g-1 | 31884 | 911 | 893 |
| soil-5g-2 | 42882 | 1335 | 1307 |
| soil-5g-3 | 45452 | 1651 | 1608 |
| soil-5g-4 | 40658 | 1858 | 1798 |
| soil-5g-5 | 29043 | 1532 | 1506 |

OTU, operational taxonomic units; S/D, non-single/double.

acceptable coverage [17]. The non-chimeric reads were a consequence of differences in DNA extraction conditions for the same soil sample. In addition, there was no significant difference in the average number of non-chimeric reads among the soil sample sizes by ANOVA; therefore, we used the obtained non-chimeric reads directly for the following analyses. The averages of relative abundances identified via homology search at the order rank for each soil sample size are shown in Fig 2. The sequence data were deposited in the DDBJ Sequence Read Archive database under the BioProject ID PRJDB10834 with BioSample IDs SAMD00260206–SAMD00236620.

## Alpha diversities of different soil sample sizes

The number of taxa at the order rank determined for different soil sample sizes is shown in Fig 3. The total number of taxa increased with decreasing soil sample size. A small soil sample size might exhibit the so-called hot spot effect; taxa with a low relative abundance have been shown to be accidentally detected owing to small and local sampling [8, 9].

Box plots of the Shannon–Wiener index across the soil sample sizes are shown in Fig 4. A large variation in the index was observed in the 5 g soil sample. Furthermore, this soil sample size included the minimum and maximum index values among all the soil sample sizes. The variation in the Shannon–Wiener index and the low efficiency of DNA extraction in the case of 5 g soil sample size could be attributed to the difficulty in disrupting cells by vertical shaking. Therefore, in the subsequent sections, we will focus only on the comparison between 0.2 and 1 g soil sample sizes and not that of the 5 g soil sample size. The results for the 5 g soil sample size are only shown in the following figures for information. Comparing the 0.2 and 1 g soil sample sizes, the determined number of taxa did not present a significant difference ($t$-test, $\alpha$ < 0.05; Fig 3). Moreover, the values of the Shannon–Wiener index did not show a significant difference (non-parametric $t$-test, $\alpha$ < 0.05; Fig 4).

## Beta diversities of different soil sample sizes

The Bray–Curtis dissimilarity index values for each soil sample size are shown in Fig 5. There was no significant difference between the 0.2 and 1 g soil sample sizes in the non-parametric

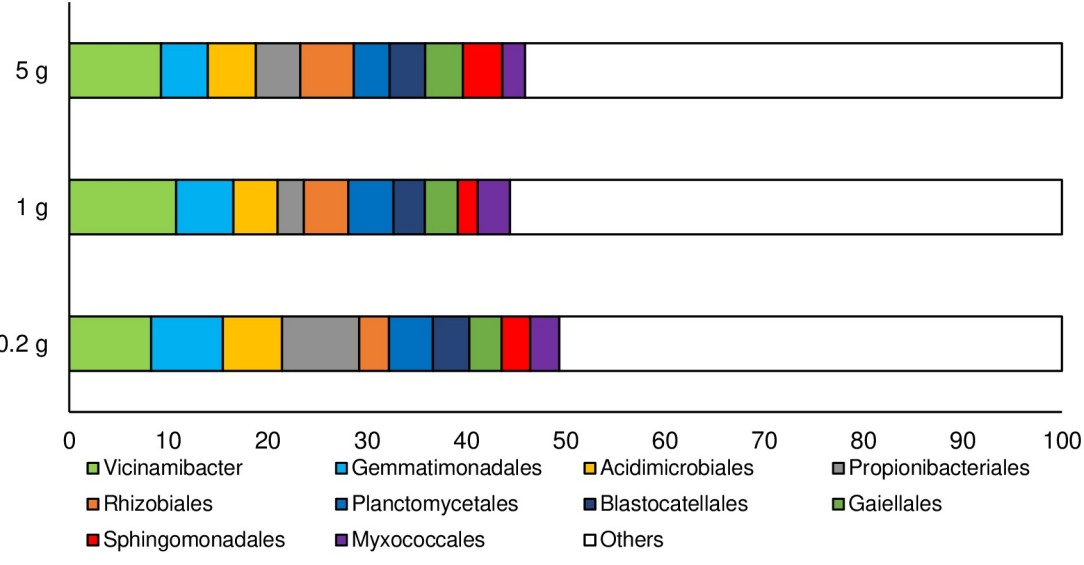

**Fig 2. Relative abundance at the order rank in each sample.** The top 10 taxa are shown. "Others" includes all minor taxa.

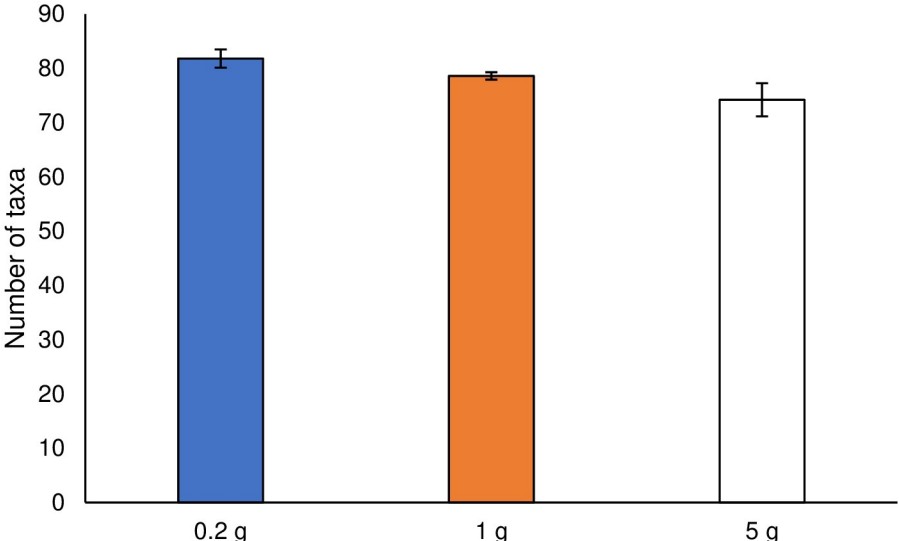

**Fig 3. Determined number of taxa at the order rank in different soil sample size.** Error bars represent standard errors.

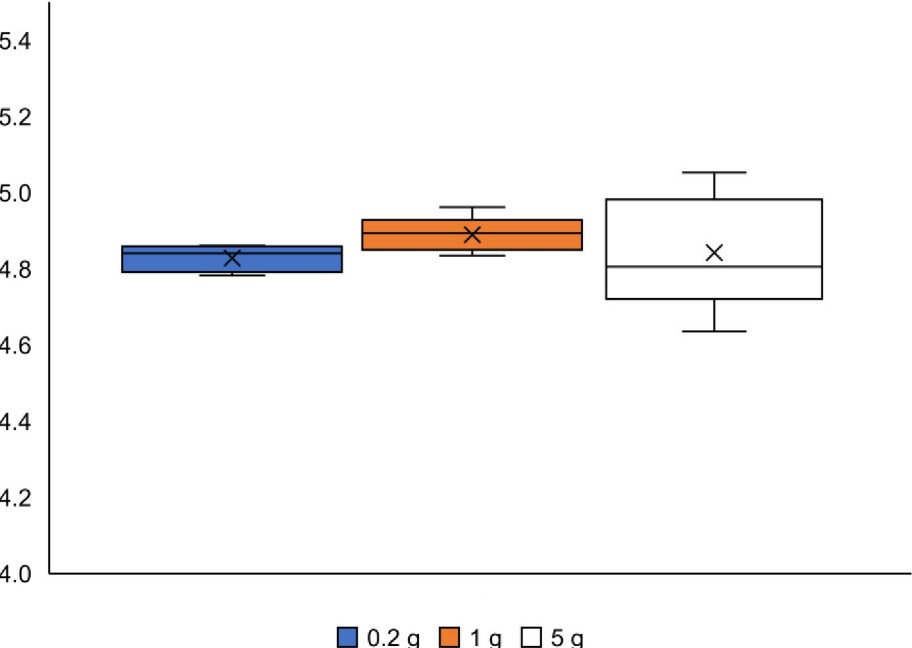

■ 0.2 g  ■ 1 g  □ 5 g

**Fig 4. Boxplots of the Shannon–Wiener indices at the order rank.** Top and bottom of the boxes are quartiles, whiskers are the maximum and minimum values, and the × marks represent the mean values. The lines that divide the box into two parts are medians (n = 5).

$t$-test ($\alpha < 0.05$). The index values of Sørensen–Dice Dissimilarity for each soil sample size are shown in Fig 6. Sørensen–Dice Dissimilarity exhibits reproducibility of detected microbial flora at every repeated sampling, depending on the presence or absence of taxa. The index value of the 0.2 g soil sample size was higher than that of the 1 g soil sample size, and

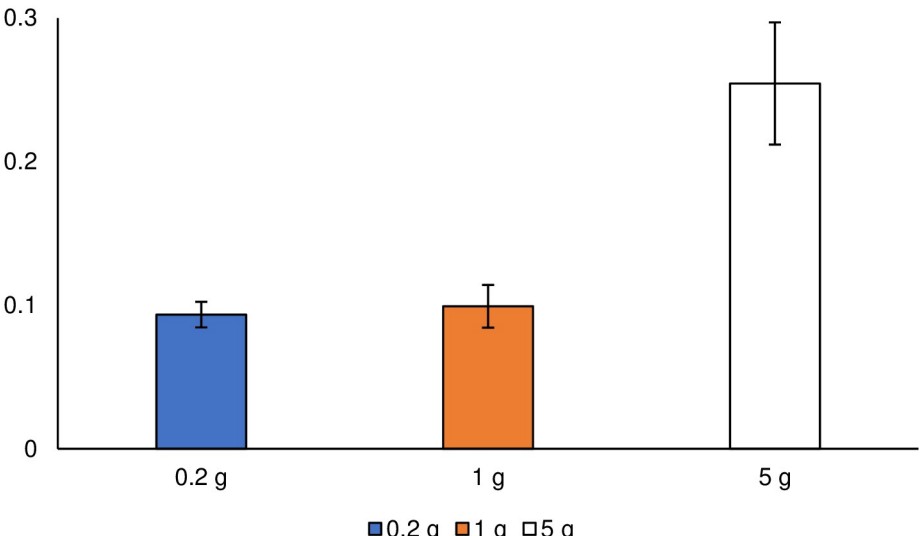

**Fig 5. Index values of Bray–Curtis dissimilarities at the order rank.** Error bars represent standard errors. There was no significant difference between soil samples of 0.2 g and 1 g.

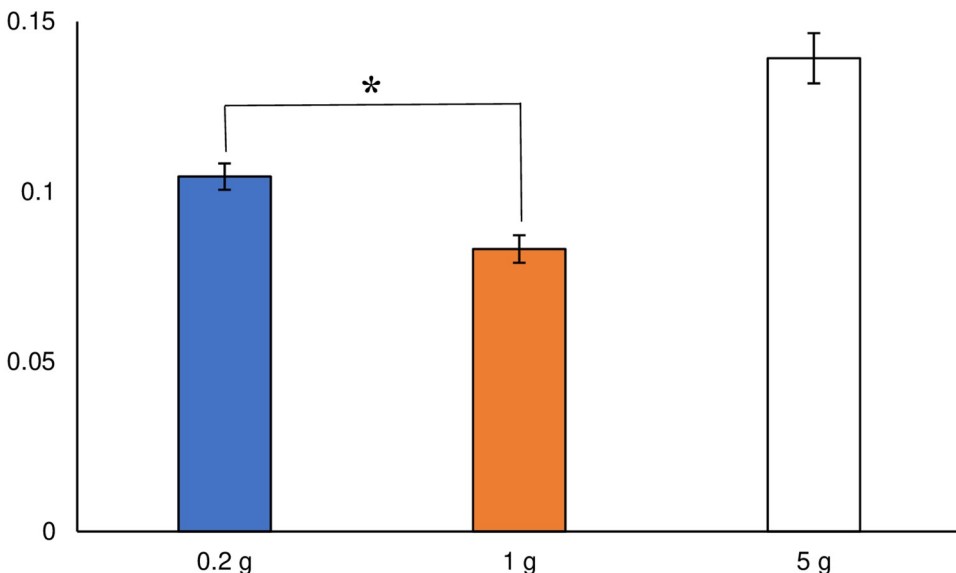

**Fig 6. Averages of the Sørensen–Dice indices at the order rank.** Different lowercase letters indicate significant differences (α < 0.05).

there was a significant difference between the two sample sizes in the non-parametric *t*-test (α < 0.05). Ellinsøe and Johnsen [9] indicated that a small soil sample size was highly susceptible to the influence of hot spots, thereby resulting in the determination of microbial compositions that fluctuate considerably. In our study, the 0.2 g soil sample size exhibited fluctuations in microbial composition. Penton et al. [10] demonstrated that a large soil sample size would be sufficient to acquire non-dominant (minor) taxa because of the abundant sequence reads provided by next-generation sequencing. The 0.2 g soil sample size, however,

detected a higher number of taxa, as well as more minor taxa (Fig 3), than the 1 g soil sample size. In our study, minor taxa were detected in the smaller soil sample size, similar to the report by Ellinsøe and Johnsen [9].

## Visualizing community structure

The NMDS plots are shown in Fig 7. Significant differences in the plot aggregation sites between the 0.2 and 1 g soil sample sizes indicate different microbial compositions. The LEfSe algorithm for the 10 major taxa, which accounted for a large proportion of the taxa, was used to clarify the effects of taxa on the determination of belonging to the two groups. The taxa with significantly different proportions between 0.2 and 1 g soil sample sizes are shown in Fig 8. Taxa of the 0.2 g soil sample size, whose relative abundance was significantly higher than that of the 1 g soil sample size, were as follows: *Acidimicrobiales*, *Propionibacteriales*, and *Sphingomonadales*. The samples of 1 g size included *Vicinamibacter* and *Rhizobiales*.

One of the common features of taxa with different relative abundances between them is the unique structure of the bacterial cell wall and cell membrane (Table 3). Gram-positive bacteria have a peptidoglycan structure, which makes them highly resistant to physical shear stress [18]. The taxa that were more abundant in the 0.2 g soil sample size (*Acidimicrobiales* and *Propionibacteriales*) belonged to gram-positive bacteria. This suggests that the bacteria in the 0.2 g soil sample size had a higher physical shear tolerance than those in the 1 g soil sample size. However, *Sphingomonadales*, belonging to gram-negative bacteria, were also identified as taxa with a larger population ratio in the case of 0.2 g soil sample size, rather than a 1 g soil sample size. *Sphingomonadales* lacks lipopolysaccharides, which are major components of the outer membrane of gram-negative bacteria. Instead, the outer membrane contains a large amount of sphingolipids [19, 20]. Sphingolipids are relatively resistant to cell disruption, which is attributed to their large capacity to form hydrogen bonds with other components of the cell membrane, resulting in a highly packed structure [20, 21]. The cell membranes of

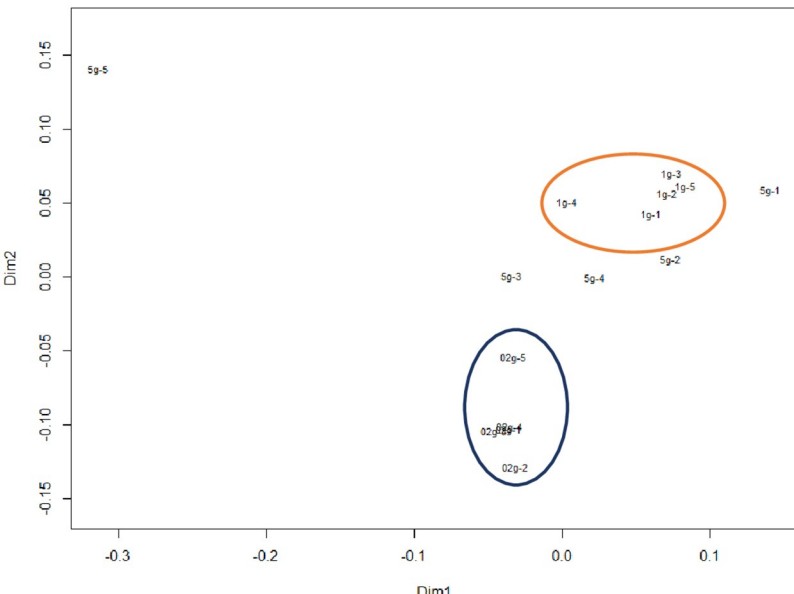

**Fig 7. Non-metric multidimensional scaling (NMDS) plots at the order rank.** The orange (upper) circle shows the aggregation sites with the 1 g soil sample sizes, and the blue (lower) circle shows those of the 0.2 g soil sample sizes. Points outside the circles are those of the 5 g soil sample sizes. 2D stress was 0.19.

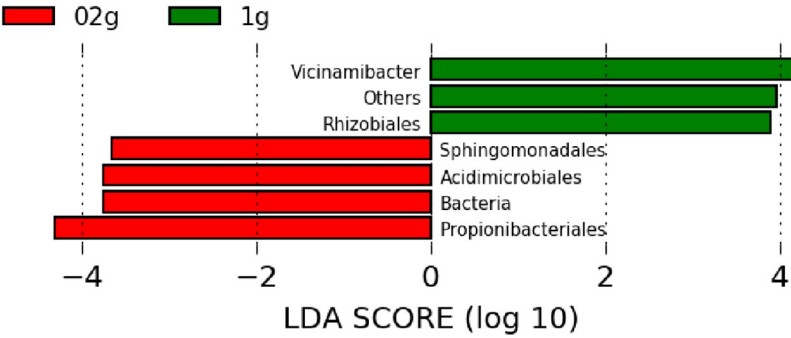

**Fig 8. Noticeable taxa at the order rank that determine the two groups.** Taxa in the 0.2 g and 1 g soil sample sizes were analyzed by the LEfSe algorithm. The 0.2 g samples presented significantly higher relative abundance of *Sphingomonadales*, *Acidimicrobiales*, and *Propionibacteriales*. 02g, 0.2 g sample size; 1g, 1 g sample size.

**Table 3. List of taxa whose relative abundances were significantly different between the 0.2 and 1 g soil sample sizes and their characteristics.**

| Kingdom | Phylum | Class | Order | Gram-Positive/Negative Bacteria | Soil sample size with a significantly large relative abundance |
|---|---|---|---|---|---|
| **Bacteria** | Acidobacteria | Vicinamibacter | Vicinamibacter | Negative | 1 g |
| **Bacteria** | Actinobacteria | Acidimicrobiia | Acidimicrobiales | Positive | 0.2 g |
| **Bacteria** | Actinobacteria | Actinobacteria | Propionibacteriales | Positive | 0.2 g |
| **Bacteria** | Proteobacteria | Alphaproteobacteria | Rhizobiales | Negative | 1 g |
| **Bacteria** | Proteobacteria | Alphaproteobacteria | Sphingomonadales | Negative | 0.2 g |

*Sphingomonadales* are denser and more tolerant to shear stress than those of other gram-negative bacteria [22]. Thus, microbial composition depends on the physical shear tolerance of the microbial community. The 0.2 g soil samples, which had a higher relative abundance of bacteria with high physical shear stress, are presumed to be exposed to severe stress conditions. This could be attributed to the small amount of soil that received higher shaking strength from the mixer. Although the difference in physical shear stress could also be derived from the type of mixer and the shape of containers, using small amounts of soil for DNA extraction, i.e., larger rated power output per unit soil weight, could provide an improved consequence with intense grinding.

## Changes in taxonomic rank

The effects of changes in the taxonomic rank on the diversity indices were compared at the order, class, and phylum ranks. The variability in microbial composition at each sample size of soil was compared using the number of taxa, Shannon–Wiener index, Bray–Curtis dissimilarity, NMDS, and Sørensen–Dice index. All indices, except for the Sørensen–Dice index, were determined at the three taxonomic ranks for the different sizes of the soil samples, and are shown in S1 and S2 Figs and Fig 9. There was no significant difference in these values ($\alpha <$ 0.05) between the 0.2 and 1 g soil sample sizes at any taxonomic rank. It could be concluded that the results of diversity assessment using these indices were stable at any taxonomic rank. The number of taxa decreased with increasing taxonomic ranks for all ranks as shown in S1 Fig. Taxa, originally classified as different in a lower hierarchy, were integrated into a common taxon by going up a rank. In this study, some minor taxa were allocated to major taxa in a

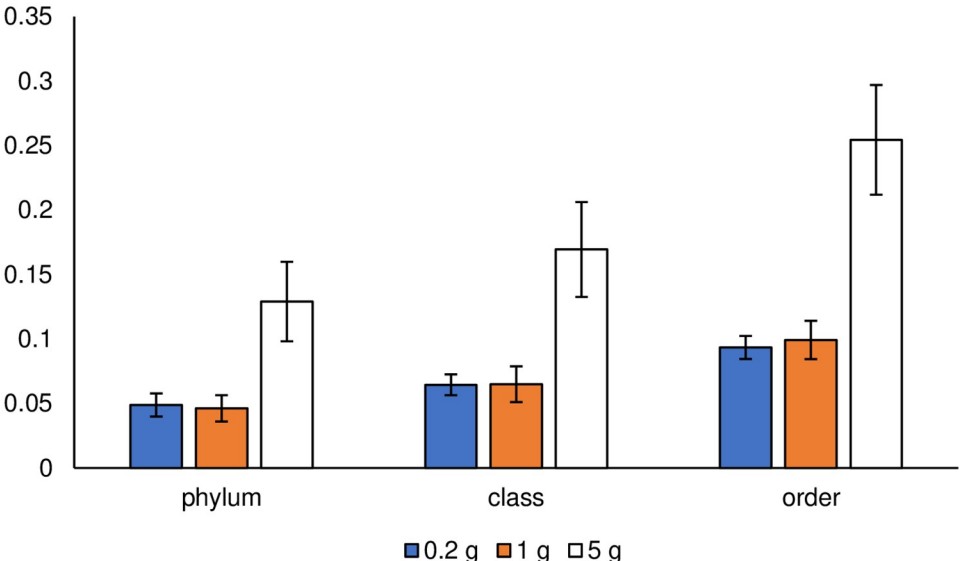

**Fig 9. Bray–Curtis dissimilarity at the three taxonomic ranks.** Error bars represent standard errors.

higher rank by integration; therefore, the discussion of microbial composition based on higher taxonomic rank might overlook the existence of minor taxa in the sample.

NMDS plots at the class and phylum ranks are shown in S3 and S4 Figs. Differences in the plot aggregation sites between the 0.2 and 1 g soil sample sizes were observed at all ranks (Fig 7, S3 and S4 Figs). In other words, in this study, the microbial community structure was preserved even when the taxonomic rank was changed.

The Sørensen–Dice indices for the three taxonomic ranks are shown in Fig 10. A non-parametric $t$-test between the index value of 0.2 and 1 g soil sample sizes revealed that the 0.2 g soil sample size was significantly smaller at the phylum rank ($\alpha < 0.05$), whereas that of the 1 g

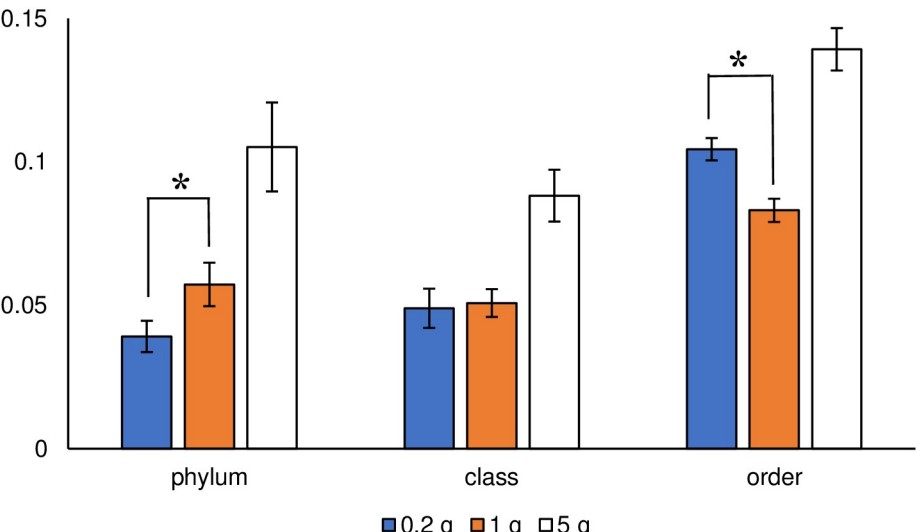

**Fig 10. Sørensen–Dice index at the three taxonomic ranks.** Error bars represent standard errors. Asterisks indicate significant differences ($\alpha < 0.05$).

soil sample size was significantly smaller at the order rank. Focusing on the detected taxa to trace the taxonomic hierarchy, some rare taxa in the 0.2 g soil sample size, which contributed to the increase in the value of the Sørensen–Dice index at the order rank, were integrated into some of the major taxa at the phylum rank. The integration might contribute to a decrease in the Sørensen–Dice index at the phylum rank. In other words, the dispersion of taxa with lower taxonomic hierarchy would occur if a wide variety of taxa sequences exist in the samples. Given the dispersion, the value of the Sørensen–Dice index of the 1 g soil sample size at order rank should be large, at least larger than that of the 0.2 g soil sample size, because the 1 g level value was larger at the higher hierarchy (phylum) rank. However, the index value was lower than that of the 0.2 g sample size at the order rank, meaning the 1 g soil sample size might not include a wide variety of taxa sequences, especially those of minor taxa. Such an assumption would contradict the current recognition that a large number of amplicon sequence reads can ensure the detection of rare species, as reported by Penton et al. [10]. One reason for this contradiction would be the taxonomic rank used in their analysis. Penton et al. [10] did not describe the taxonomic rank they used; however, if they used a high taxonomic rank, such a higher taxonomic hierarchy, it might not be appropriate for the detection and discussion of rare species because of the characteristic of integration. When considering a microbial community structure at lower taxonomic ranks, such as species, a large number of amplicon sequence reads may not ensure the detection of minor taxa due to a wide variety of branching in the phylogenetic tree. In this case, applying a small soil sample size might effectively detect minor taxa using the hot spot effect. In their review, Gołębiewski and Tretyn [23] summarized that small soil sample sizes (e.g., 0.25 g) are appropriate for studying microbial diversity.

## Determination of soil sample size

In the previous sections, two soil sample sizes, 0.2 and 1 g, were weighed for the metagenomic analysis of soil microbial community structure. Although both sample sizes had advantages and disadvantages, we concluded that a 0.2 g soil sample size is recommended, especially for practical use. Regarding the 0.2 g soil sample size, a wide variety of taxa, including those with resistance to shear stress, were detected (Fig 2), possibly because a vortex mixer gave a higher rated power output per unit soil weight, and the smaller soil sample size was more sensitive to the hot spot effect. The values of the diversity indices, except for the Sørensen–Dice index, were almost the same between the 0.2 and 1 g soil sample sizes at different taxonomic ranks. In addition, the 0.2 g soil sample size requires smaller amounts of reagents and consumable goods for DNA extraction, making the operation economical. In contrast, the 1 g soil sample size presented stable microbial community structures at the lower taxonomic rank (order rank, Fig 10). The DNA extraction efficiency was better than that of the 0.2 g soil sample size (Fig 1). With the continued improvements in the performance of taxonomic databases and analytical techniques, an analysis of microbial community structure at a lower taxonomic rank, such as species rank, would be possible. In such a case, under the possible dispersion of taxa with the lowering of the taxonomic hierarchy, presenting a stable structure would also be required, and a larger soil sample size would be appropriate in that case.

In addition to soil sample size, the relationship between the amount of soil (soil sample size) and extraction container volume (tube volume) would be important for DNA extraction and subsequent analyses. Penton et al. [10] also demonstrated DNA extraction and subsequent analyses using a soil sample size of 0.25 g with a different DNA extraction kit (2 mL tube). The kit was prepared by the same manufacturer as that used in the cases of the 1–10 g soil sample sizes (50 mL tube). When comparing the 0.25 and 1 g soil sample sizes on numbers of bacterial OTUs determined, the 0.25 g soil sample size provided stable analytical results, such as

reproducibility of microbial community structure. This does not indicate that a large soil sample size is always better. It only implies that the relationship between the soil sample size and extraction container volume is also considered essential.

## Conclusions

In this study, we prepared soil samples 0.2, 1, and 5 g in size, and evaluated the effect of the soil sample size on DNA extraction and microbial community structure analysis using amplicon sequence data. We focused on the relationship between the soil amount and the extraction container volume in DNA extraction, and the differences in microbial community structure between two soil sample sizes (0.2 g and 1 g) at different taxonomic ranks. In the 0.2 and 1 g soil sample sizes, the numbers of taxa and α- and β-diversity indices used were stable, with almost the same values at each taxonomic rank. However, NMDS showed that the microbial compositions of the two cases at the order rank were clearly different, and the difference was also observed at higher taxonomic ranks. The soil sample size and tube volume affected the estimated microbial community structure. A 0.2 g soil sample size would be preferable to larger sample sizes because of the possible higher shearing force for DNA extraction and lower costs for practical use with smaller amounts of consumables. When the taxonomic rank was changed from order to phylum, some minor taxa identified at the order rank were integrated into major taxa at the phylum rank. The integration affected the value of the Sørensen–Dice index. Therefore, the microbial community structure analysis was influenced by the taxonomic rank applied.

## Supporting information

**S1 Fig. Number of taxa determined at the three taxonomic ranks.** Error bars represent standard errors.
(TIF)

**S2 Fig. Shannon–Wiener index at the three taxonomic ranks.** Error bars represent standard errors.
(TIF)

**S3 Fig. Non-metric multidimensional scaling (NMDS) plot at the class rank.** Aggregation points of the 0.2 g soil sample size are shown in blue (right) circle and those of the 1 g soil sample size are shown in orange (left) circle. Points outside the circles are for the 5 g soil sample size. 2D stress was 0.14.
(TIF)

**S4 Fig. Non-metric multidimensional scaling (NMDS) plot at the phylum rank.** Agglomeration points of the 0.2 g soil sample size are shown in the blue (left) circle, and those of the 1 g soil sample size are shown in the orange (right) circle. Points outside the circles are for the 5 g soil sample size. 2D stress was 0.061.
(TIF)

## Acknowledgments

We would like to thank Editage (www.editage.com) for English language editing.

## Author Contributions

**Conceptualization:** Hiroki Morita, Satoshi Akao.

**Data curation:** Hiroki Morita.

**Formal analysis:** Hiroki Morita.

**Funding acquisition:** Satoshi Akao.

**Investigation:** Hiroki Morita.

**Methodology:** Hiroki Morita, Satoshi Akao.

**Project administration:** Satoshi Akao.

**Resources:** Satoshi Akao.

**Validation:** Satoshi Akao.

**Visualization:** Hiroki Morita.

**Writing – original draft:** Hiroki Morita.

**Writing – review & editing:** Satoshi Akao.

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
