## [Decision Letter · Decision Letter 0]

4 Aug 2021

PONE-D-21-20731

The effect of soil sample size, for practical DNA extraction, on soil microbial diversity in different taxonomic ranks

PLOS ONE

Dear Dr. %Satoshi%,

Thank you for submitting your manuscript to PLOS ONE. After careful consideration, we feel that it has merit but does not fully meet PLOS ONE’s publication criteria as it currently stands. Therefore, we invite you to submit a revised version of the manuscript that addresses the points raised during the review process.

ACADEMIC EDITOR: Ashwani Kumar

We look forward to receiving your revised manuscript.

Kind regards,

Ashwani Kumar

Academic Editor

PLOS ONE

Journal Requirements:

[NO]. 

4. We note that S-Figures 1 and 2 in your submission contain copyrighted images. All PLOS content is published under the Creative Commons Attribution License (CC BY 4.0), which means that the manuscript, images, and Supporting Information files will be freely available online, and any third party is permitted to access, download, copy, distribute, and use these materials in any way, even commercially, with proper attribution. For more information, see our copyright guidelines: http://journals.plos.org/plosone/s/licenses-and-copyright.

a) You may seek permission from the original copyright holder of Figures S-Figure 1 and S-Figure 2 to publish the content specifically under the CC BY 4.0 license. 

Reviewers' comments:

Reviewer's Responses to Questions

**Comments to the Author**

1. Is the manuscript technically sound, and do the data support the conclusions?

Reviewer #1: Yes

Reviewer #2: Yes

Reviewer #3: Partly

2. Has the statistical analysis been performed appropriately and rigorously? 

Reviewer #1: Yes

Reviewer #2: Yes

Reviewer #3: Yes

3. Have the authors made all data underlying the findings in their manuscript fully available?

Reviewer #1: Yes

Reviewer #2: Yes

Reviewer #3: Yes

4. Is the manuscript presented in an intelligible fashion and written in standard English?

Reviewer #1: Yes

Reviewer #2: Yes

Reviewer #3: Yes

5. Review Comments to the Author

Reviewer #1: your work "the effect of soil sample size, for practical DNA extraction, on soil microbial diversity in different taxonomic ranks" is very interesting and useful for cauterising soil and understanding the nature of microorganisms presentation.

Reviewer #2: The manuscript entitled ‘The effect of soil sample size, for practical DNA extraction, on soil microbial diversity in different taxonomic ranks' tried to determine the optimal soil sample size in regard for microbial community structure. They focused on the relationship between soil amount and DNA extraction container volume and the alteration in microbial composition at different taxonomic ranks (order, class, and phylum). Three soil sample size were tested, and they concluded that microbial diversity and diversity assessment were stable on 0.2 and 1-g samples. It is interesting to found the effect of sample size to microbial diversity. Thus, I think it is valuable to publish the paper, which would provide technical insight of sampling size.

However, I would like to propose a small suggestion to the author to adjustment the conclusion. It would be easier for the reader if the conclusion is brief and concise, and which sample size practice is the best.

Reviewer #3: General comments

This manuscript reports on the optimal soil sample size for microbial community structure analysis. Besides, this study also focused on the relationship between soil amount and DNA extraction container volume and the alteration in the microbial composition at different taxonomic ranks. The results suggest that microbial community structure analysis is influenced by the taxonomic rank applied.

However, I have some concerns about some of the methodology used in this study. For example, the sample sizes used and also the number of genes used in the microbial composition analysis. These indirectly impacts the general results and findings of this study. Moreover, the authors need to clarify the novelty of their study as the same study has been shown in other research, Penton et al., 2016, even though it has been cited in the study. The authors should make this clear in the introduction and make a more comprehensive conclusion.

Hence, I have to reject this manuscript because it does not meet the PLoS ONE criteria

However, these studies still have an impact of results in its own way

* Consider carefully proofreading – there are several grammatical errors and also replicate of a sentence.

6. PLOS authors have the option to publish the peer review history of their article (what does this mean?). If published, this will include your full peer review and any attached files.

Reviewer #1: No

Reviewer #2: No

Reviewer #3: No

---

## [Author Response · Author response to Decision Letter 0]

8 Sep 2021

Reviewer #1: your work "the effect of soil sample size, for practical DNA extraction, on soil microbial diversity in different taxonomic ranks" is very interesting and useful for cauterising soil and understanding the nature of microorganisms presentation.

Author: Thank you very much for your encouraging words.

Reviewer #2-1: The manuscript entitled ‘The effect of soil sample size, for practical DNA extraction, on soil microbial diversity in different taxonomic ranks' tried to determine the optimal soil sample size in regard for microbial community structure. They focused on the relationship between soil amount and DNA extraction container volume and the alteration in microbial composition at different taxonomic ranks (order, class, and phylum). Three soil sample size were tested, and they concluded that microbial diversity and diversity assessment were stable on 0.2 and 1-g samples. It is interesting to found the effect of sample size to microbial diversity. Thus, I think it is valuable to publish the paper, which would provide technical insight of sampling size.

Author:We appreciate the reviewer’s high evaluation.

Reviewer #2-2: However, I would like to propose a small suggestion to the author to adjustment the conclusion. It would be easier for the reader if the conclusion is brief and concise, and which sample size practice is the best.

Author: Thank you very much for the useful suggestion. We have concluded that the 0.2-g soil sample level would be preferable due to its higher detection of bacteria with a high tolerance of shearing force and its economic efficiency [L. 31 and 402]. Also, we made the Abstract and the Conclusion briefer and more concise [L. 17, 27, 30, 393, and 401]. 

Reviewer #3-1: General comments

This manuscript reports on the optimal soil sample size for microbial community structure analysis. Besides, this study also focused on the relationship between soil amount and DNA extraction container volume and the alteration in the microbial composition at different taxonomic ranks. The results suggest that microbial community structure analysis is influenced by the taxonomic rank applied.

However, I have some concerns about some of the methodology used in this study. For example, the sample sizes used and also the number of genes used in the microbial composition analysis. These indirectly impacts the general results and findings of this study. 

Author: Thank you very much for your meaningful comments. As the reviewer mentioned, the soil sample size might affect the microbial composition analysis. Also, we have focused on the relationship between the soil sample size and the DNA extraction conditions (e.g., the volume of the container). Regarding the study using amplicon sequence data, Penton et al. and Song et al. have studied the soil sample size effect. However, However, Penton et al. evaluated the effect of the soil sample sizes (1, 5, and 10 g) using the DNA extraction kit for 10 g of soil, with the result that a larger soil sample size would be preferable. In this respect, each DNA extraction kit would have an optimal soil volume, and the smaller soil sample size in the study might deviate from the optimal volume. In other words, the relationship between the soil volume and the container volume for DNA extraction would be important, as well as the soil sample size effect, and we have investigated the points. Song et al. also elucidated the DNA extraction efficiency using the different soil sample sizes (0.25 g and 10 g) with each optimal-size DNA extraction kit, respectively. There was no significant difference in the OUT richness between the two soil sample sizes, and this result would support our idea, meaning the relationship between the soil volume and the container volume is also essential. Therefore, we added the report by Song et al. in the reference list [L. 71].

We would have liked to carry out further soil sample size cases. However, the horizontal shaking adapter for 50 mL tube volume for the vortex mixer can only equip two tubes. It is inconvenient for practical use because we usually treat many samples at one time. Therefore, we have tried a vertical, not a horizontal shaker, which can equip 6 tubes at once, to extract DNA for practical use.

We would think that the sample size indicated by Reviewer#3 means the soil sample size. However, generally, sample size means repetition of one sample. Therefore, to distinguish soil sample volume from the so-called sample size, we added the word “soil” prior to sample size, meaning soil sample volume. Also, we apologize for the confusion. Relating to the so-called sample size, we also added the reason for our repetition number of one sample (n = 5) by referring to Song et al. (2014) and Vestergaard et al. (2017) [L.50 and 117].

The number of genes also would affect microbial composition analyses. We think the reviewer’s indication has two meanings: one is the volume of sequence reads, and the other is the equality of the sequence reads by rarefaction. For the former points, 30,000-40,000 reads were obtained through the experimental protocols in our study. Therefore, the sequence reads would be enough to evaluate microbial composition analysis (Schöler et al., 2017). Also, we rewrote the sequence reads of Penton et al. from its total reads (over 5,000,000 reads) to the reads of each sample (over 56,000 reads) for the comparison of our sequence reads [L. 69]. For the latter points, we used the same soil, not the different soil. So, the number of sequence reads was one of the consequences of comparing the different soil sample sizes for DNA extraction. Therefore, we did not rarefy the read number and conducted the microbial community analysis using the obtained read number. We mentioned the point in the revised manuscript [L. 193]. By the way, the average number of sequence reads in each soil sample size would be confirmed to be almost the same (no significant difference) by applying ANOVA [L. 161]. 

Reviewer #3-2: Moreover, the authors need to clarify the novelty of their study as the same study has been shown in other research, Penton et al., 2016, even though it has been cited in the study. The authors should make this clear in the introduction and make a more comprehensive conclusion.

Author: Thank you for your advice. Our main discussion points are as follows; 1) effects of the target taxonomic rank and annotated taxa in microbial diversity analysis and 2) the relationship between soil volume and tube volume for DNA extraction. Firstly, we summarized the section of our motivations in an orderly way (firstly, secondly, finally) in the Introduction section [L. 76, 88, and 98]. Then, we noticed that the explanation of the former discussion point, especially using annotated taxa for microbial diversity analysis, had not been adequate in the original manuscript, so we described the point in more detail [L. 73 and 108]. We added concerns about using OTUs not annotated taxa for microbial diversity analysis [L. 76]. Also, we referred to the report by Song et al. (2015) to support our discussion [L. 71, 95]. By the way, we found our misunderstanding about the target taxonomic rank in the study by Penten et al. (2016). They did not determine a specific taxonomic rank. We apologize and corrected the point [L. 76].

We also discussed the importance of setting target taxonomic rank and using annotated taxa in microbial diversity analysis in the Results and Discussion section. We added some ideas for the transition of the Sørensen-Dice index with lowering the taxonomic rank relating to the detection of minor taxa. In addition, we wrote the possible reason for the different conclusions about detecting minor taxa between ours and Penton et al.’s. [L.339 and 348]. We also added the reference that discussed soil sample size for the detection of minor taxa [L. 355].

We re-evaluated our results for a more straightforward conclusion and concluded that a 0.2-g soil sample size would be better than a 1-g soil sample size because the soil sample size could detect taxa with higher physical shear tolerance and was carried out at a lower cost. We rewrote the choice of the two soil sample sizes in the Results and Discussion section (Determination of Sample Size) [L. 364] and the Conclusions section [L. 402]. In the Determination of Sample Size section, we summarized the advantages of each soil sample size (0.2 and 1 g) [L. 363].

We added the following contents to explain the points mentioned (underline shows the added sentences and phrases). In addition, we cleared some information in the Introduction that did not relate to the main discussing points to avoid redundancy [L. 43]. Also, we summarized the Conclusion to make it more concise [L. 393].

Reviewer #3-3: Hence, I have to reject this manuscript because it does not meet the PLoS ONE criteria

However, these studies still have an impact of results in its own way

* Consider carefully proofreading – there are several grammatical errors and also replicate of a sentence.

Author: Thank you for your advice. We have again discussed our novelty and the differences from previous reports and revised the manuscript as we mentioned above. We have sincerely read the revised manuscript and used the English proofreading service again.

---

## [Editor Report · Decision Letter 1]

3 Nov 2021

The effect of soil sample size, for practical DNA extraction, on soil microbial diversity in different taxonomic ranks

PONE-D-21-20731R1

Dear Dr. Akao,

We’re pleased to inform you that your manuscript has been judged scientifically suitable for publication and will be formally accepted for publication once it meets all outstanding technical requirements.

Kind regards,

Ashwani Kumar

Academic Editor

PLOS ONE
---

## [Editor Report · Acceptance letter]

8 Nov 2021

PONE-D-21-20731R1 

The effect of soil sample size, for practical DNA extraction, on soil microbial diversity in different taxonomic ranks 

Dear Dr. Akao:

I'm pleased to inform you that your manuscript has been deemed suitable for publication in PLOS ONE. Congratulations! Your manuscript is now with our production department. 

Kind regards, 

on behalf of

Dr. Ashwani Kumar 

Academic Editor

PLOS ONE